# *Euonymus sachalinensis* Induces Apoptosis by Inhibiting the Expression of c-Myc in Colon Cancer Cells

**DOI:** 10.3390/molecules28083473

**Published:** 2023-04-14

**Authors:** So-Mi Park, Wona Jee, Ye-Rin Park, Hyungsuk Kim, Yun-Cheol Na, Ji Hoon Jung, Hyeung-Jin Jang

**Affiliations:** 1College of Korean Medicine, Kyung Hee University, 26 Kyungheedae-ro, Dongdaemun-gu, Seoul 02447, Republic of Korea; psm991030@naver.com (S.-M.P.); jee1ah@khu.ac.kr (W.J.); dutndi@naver.com (Y.-R.P.); johnsperfume@gmail.com (J.H.J.); 2Department of Science in Korean Medicine, Graduate School, Kyung Hee University, Seoul 02447, Republic of Korea; 3Department of Korean Rehabilitation Medicine, Kyung Hee University Medical Center, Seoul 02447, Republic of Korea; kim0874@hanmail.net; 4Western Seoul Center, Korea Basic Science Institute, 150 Bugahyeon-ro, Seodaemun-gu, Seoul 03759, Republic of Korea; nyc@kbsi.re.kr

**Keywords:** *Euonymus sachalinensis*, colon cancer, c-Myc, apoptosis, oncogene, anti-cancer

## Abstract

We hypothesized that *Euonymus sachalinensis* (ES) induces apoptosis by inhibiting the expression of c-Myc in colon cancer cells, and this study proved that the methanol extract of ES has anticancer effects in colon cancer cells. ES belongs to the Celastraceae family and is well known for its medicinal properties. Extracts of species belonging to this family have been used to treat diverse diseases, including rheumatoid arthritis, chronic nephritis, allergic conjunctivitis, rhinitis, and asthma. However, ES has been targeted because there are currently few studies on the efficacy of ES for various diseases, including cancer. ES lowers cell viability in colon cancer cells and reduces the expression of c-Myc protein. We confirm that the protein level of apoptotic factors such as PARP and Caspase 3 decrease when ES is treated with Western blot, and confirm that DNA fragments occur through TUNEL assay. In addition, it is confirmed that the protein level of oncogenes CNOT2 and MID1IP1 decrease when ES is treated. We have also found that ES enhances the chemo-sensitivity of 5-FU in 5-FU-resistant cells. Therefore, we confirm that ES has anticancer effects by inducing apoptotic cell death and regulating the oncogenes CNOT2 and MID1IP1, suggesting its potential for use in the treatment of colon cancer.

## 1. Introduction

Cancer is a disease with a high mortality rate because cell growth is not controlled [1]. Among them, colon cancer, which begins in the mucous membrane of the large intestine, is the second deadliest cancer after lung cancer, with 9.4% of all 9.9 million cancer deaths, according to the World Health Organization’s 2020 statistics [2]. Colon cancer is common in both men and women [3], and the incidence rate is known to be increasing in adults under the age of 50 [4]. Major causes of colon cancer include aging, smoking, drinking, obesity, and heredity, and preventive methods include positive lifestyle changes such as exercise and hormone replacement therapy. Major chemotherapy includes 5-Fluorouracil (5-FU) and leucovorin, and treatment using chemotherapy has attracted attention as an effective treatment for colon cancer [5]. Fortunately, recent information has shown that the incidence rate in humans over 65 years old is gradually decreasing, but about 50% of colon cancer in such patients recur and various side effects can occur due to strong toxicity when treated with chemotherapy [6]. Therefore, the development of colon cancer treatments that use natural products with relatively low toxicity has attracted attention [7].

The plants of genus Euonymus has been used as a treatment for several diseases and is known to have traditional medicinal properties [8]. For example, *Euonymus alatus* has been found to be a potential treatment candidate for cancer, hyperglycemia, dysmenorrhea, and diabetes complications in some Asian countries [9]. *Euonymus alatus* Sieb. is widely used in folk medicine to treat various cancers such as cancer of the liver, esophagus, stomach, and uterus [10]. In addition, *Euonymus carnosus* [11] and *Euonymus oblongifolius* [12] were found to inhibit lipopolysaccharide (LPS)-induced nitric oxide (NO) production in the murine microglial BV2 cell line. They have also been found to inhibit cytotoxic activity in several human cancer cell lines. *Euonymus sachalinensis* (ES) (F. Schmidt) Maxim., which belongs to the Celastraceae family that we used in this study, is mainly distributed in Southeast Asia, Sakhalin Island, and Hungary, and has been used as a decorative shrub [13]. Traditional indications of ES include the treatment of stomachalgia [14], and its chemical components of ES mainly comprise acanfolioside [15], azadirachtin [15], and evonolin (4-deoxyevonine) [13]. In addition to ES, extracts traditionally used in stomach trouble include *Gladiolus dalenii* van Geel [16] and *Morus mesozygia* [17], and studies on their efficacy have been conducted. However, research on ES is currently limited to the extent mentioned earlier. As such, research on the effectiveness of ES used in treatment is currently insufficient, so we studied to reveal the effectiveness of ES in colon cancer cells.

c-Myc belongs to the *MYC* gene family and is one of the genes with neoplastic potential among this family [18]. c-Myc is an important transcription factor that can be tightly regulated in normal cells but is overexpressed in cancer cells and plays multiple roles [19]. Specifically, c-Myc, a proto-oncogene, is involved in cell growth, proliferation, the cell cycle, metabolism, and apoptosis in cancer cells [20]. It has now been found that c-Myc is amplified in various types of cancer, such as colonic carcinoma [21], breast carcinoma [22,23], and lung cancer [24]. In particular, c-Myc overexpression has been confirmed in approximately one in three breast and colon carcinomas [18]. Considering that c-Myc plays a pivotal role in cancer, it is an ideal target for cancer treatment [25]. Therefore, we considered the expression of c-Myc with these characteristics as an important factor in this study.

In addition, we investigated the expression of oncogenes, such as midline1 interacting protein 1 (MID1IP1), which is regulated by the carbohydrate response element binding protein, and CCR4-NOT2 (CNOT2), which is involved in adipogenic activity [26]. MID1IP1 overexpression activates c-Myc, and inhibition of CNOT2 enhances its antitumor effect by inducing apoptosis in colorectal cancer cells through MID1IP1 [27]. In other words, inhibition of CNOT2 in HCT116 cells not only decreased c-Myc, but also decreased the apoptosis-associated protein pro-Caspase 3, known as an executioner Caspase in apoptosis, and the pro-forms of PARP, poly adenosine diphosphate-ribose polymerase [26]. It was confirmed that as pro-PARP and pro-Caspase 3 decreased, cleaved-PARP and cleaved-Caspase 3 were adjusted upward, leading to apoptosis. In addition, after making 5-FU-resistant HCT116 cells, we found that ES also acts on 5-FU-resistant HCT116 (5-FU-R-HCT116) cells.

In this study, we evaluated whether ES was effective in the treatment of colon cancer by treating a colon cancer cell line with ES and checking the expression change of c-Myc. We also demonstrated that apoptosis was induced by the treatment of colon cancer cell lines with ES and that ES was effective in HCT116 cells resistant to 5-FU. Therefore, our data revealed that ES is effective in the treatment of colon cancer and deserves further study.

## 2. Results

### 2.1. ES Inhibits Cell Viability in Colon Cancer Cells

We investigated whether ES inhibits cell viability in colon cancer cells using the MTT assay. When the cells were treated with ES at different concentrations, the cell viability of colon cancer cells (HCT116^p53+/+^, HCT116 ^p53−/−^ and HT29, DLD-1 cells) decreased in a dose-dependent manner (Figure 1B). These results suggest that ES inhibited the cytotoxicity of colon cancer cells.

### 2.2. ES Decreases c-Myc Expression and Induces Apoptosis in Colon Cancer Cells

We found that ES inhibited c-Myc and the protein expression level of apoptotic factors, including pro-PARP, cleaved-PARP, pro-Caspase 3, cleaved-Caspase 3 in a dose-dependent (Figure 2A) and time-dependent manner (Figure 2B). Furthermore, we confirmed changes in expression of c-Myc when ES was treated on cells by immunofluorescence. As shown in Figure 2C, nuclear c-Myc levels were decreased by ES treatment in colon cancer cells. Furthermore, we confirmed whether apoptosis was the cause of colon cancer cell viability after ES treatment. The number of TUNEL positive cells increased as a result of the TUNEL assay. In addition, DNA fragmentation occurred in the group treated with ES 100 µg/mL (Figure 2D).

### 2.3. ES Decreases the Expression of Oncogenes in Colon Cancer Cells

We identified changes in protein expression levels of oncogenes such as CNOT2 and MID1IP1 when ES was treated in colon cancer cells through Western blotting. The results confirmed that both the dose-dependent manner (Figure 3A) and the time-dependent manner (Figure 3B) gradually decreased.

### 2.4. ES Regulates c-Myc Expression by Serum Stimulation

First, the cell cycle was synchronized using 0.2% and 20% FBS. The expression level of c-Myc was confirmed when the cells were treated with ES. Since c-Myc is characterized by a rapid response to serum stimulation, we investigated whether ES affects the induction of serum reactivity in colon cancer cells by altering the expression of c-Myc. The control group was treated with DMSO. In the group treated with ES, the expression of c-Myc was lower at 12 and 24 h than at 6 h (Figure 4). These results suggest that ES treatment reduces the serum-stimulated expression of c-Myc.

### 2.5. ES Reduces c-Myc Protein Stability in Colon Cancer Cells

We evaluated the stability of c-Myc using CHX. After 24 h of ES treated in each colon cancer cell, it was exposed to CHX for 0, 30, 60, and 90 m. As a result, the half-life of c-Myc was lower in the ES group than in the control group (treated with DMSO) (Figure 5). These results suggest that ES inhibits the stability of the c-Myc protein.

### 2.6. ES Acts in HCT116 and 5-FU-R-HCT116 Cells

p53, a tumor suppressor, plays a role in limiting cell proliferation by inducing apoptosis [28]. 5-FU treatment increases the translation, stability, and transcriptional activity of p53, which is known as a sequence-specific transcription factor, and the target genes of p53 promote anti-cancer processes, such as apoptosis [29]. Therefore, we confirmed that the expression level of p53 did not change through Western blotting after preparing cells resistant to 5-FU. We confirmed that the cells were resistant to 5-FU by verifying that there was no difference in the expression of p53 compared to normal HCT116 cells (Figure 6A). When parental HCT116 and 5-FU-R-HCT116 cells were treated with ES, the cell viability and c-Myc expression decreased in all cells (Figure 6B,C).

### 2.7. The Methanol Extract of ES by LC-MS Analysis

LC-MS analysis was used to investigate the separation of components in ES. Through this, the main components of the anticancer effect of ES were identified in colon cancer cells. Most of the ingredients were recognized in ESI+ and are as follows: ESI+: [M+H_2_O]+, [M+K]+. It was also used to qualitatively analyze LC-MS data. Two peaks were detected in the LC-MS chromatogram of the ES extract, *x*-axis means the retention time of the peak, and *y*-axis means the intensity of the peak (Figure 7). The compounds detected through LC-MS analysis are summarized in Table 1.

### 2.8. Schematic Diagram of the Anti-Colon Cancer Activity of ES

The previous data show that ES induces apoptosis in colon cancer cells. Finally, we summarized the anti-colon cancer effects of ES in vitro by a mechanism scheme (Figure 8).

## 3. Discussion

There is currently no cure for colon cancer, and the treatments available are associated with a number of side effects. Therefore, it is necessary to discover treatment candidates that are effective for colon cancer treatment and have relatively few side effects. The current study demonstrated that ES could be a therapeutic candidate for colon cancer. We investigated whether ES causes cell apoptosis, focusing on c-Myc expression in HCT116 cells^p53+/+^ and HT29 cells among several colon canceler cell lines. Therefore, HCT116^p53+/+^ cells are simply marked as HCT116 except for in Figure 1. Our results confirmed that ES induced apoptosis by reducing the expression of c-Myc and checking cell viability and protein levels in HCT116 and HT29 cells. ES also regulates the expression levels of c-Myc, pro-PARP, cleaved-PARP, pro-Caspase 3, and cleaved-Caspase 3, which are apoptosis-related genes, and CNOT2 and MID1IP1, which act as oncogenes.

Apoptosis, a type of cellular suicide, is involved in the development of cells and tissues through mitochondrial or endogenous pathways and in various physiological and pathological conditions, which usually activate caspase protein degradation [30]. c-Myc, known as a proto-oncogene, plays a role in various processes, such as cell apoptosis and senescence, as well as in physiological functions, such as cell-cycle control, metabolism, and protein biosynthesis [31]. In addition, although it has diverse functions, this study focused on the involvement of c-Myc in apoptosis, providing evidence that ES has anti-cancer effects. Although c-Myc is tightly controlled in normal cells, it is activated in most human tumors, particularly cancer [19]. When c-Myc is activated, it is involved in cancer cell growth through the transcription and replication of DNA and regulation of stemness and differentiation of cancer [31]. Previous studies have shown that c-Myc plays a role in the development of colon cancer [2].

Caspase 3 is a representative factor that can indicate the occurrence of apoptosis [32]. This also enables the activation of cascade [33]. When Caspase 3 is activated, pro-PARP is cleaved to form cleaved-PARP, which is a representative factor in apoptosis [34]. PARP plays an important role in activating DNA recovery pathways [35]. Our Western blot results showed a downregulation of pro-PARP and pro-Caspase 3 and an upregulation of cleaved-PARP and cleaved-Caspase 3 when colon cancer cells were treated with ES. In addition, Western blotting demonstrated that the expression of MID1IP1 was reduced when the colon cancer cells were treated with ES. Previous studies have revealed that CNOT2 is related to MID1IP1, which is associated with the regulation of c-Myc expression and apoptosis in cancer cells, and that CNOT2 knockdown induces apoptosis through MID1IP1 [26]. Therefore, it appears that ES induces apoptosis by regulating the expression of the aforementioned proteins in colon cancer cells.

Based on the current study, it appears that ES has the potential to be used as an anti-cancer drug for the treatment of colon cancer. We found that when colon cancer cells were treated with ES, cell viability inhibition (by MTT assay), the expression of c-Myc (by Western blotting, serum stimulation, IF, TUNEL assay), apoptotic proteins, oncogene inhibition (by Western blotting), and c-Myc stability reduction (by CHX) were apparent. In addition, studies have been conducted on 5-FU, which regulates p53 and which also plays an important role in the anticancer process; it is used in the treatment of metastatic colon cancer, but has several problems related to resistance. We confirmed that ES also works in 5-FU-resistant cells by confirming a decrease in the expression of c-Myc in HCT116^p53+/+^ cells and 5-FU-resistant HCT116^p53+/+^ cells through Western blotting [36]. In other words, since a new perspective on ES has been highlighted in this study, it is necessary to conduct additional component analyses and evaluate the efficacy of ES.

ES was analyzed by LC-MS in this study because it is necessary to identify compounds to evaluate the effectiveness of natural products. As a result, two components of ES that affect the colon cancer cell were identified. Evonine [37] and Acanfolioside [15] have been detected as active substances in colon cancer cells. Celastraceae alkaloids, Evonine was isolated from *Euonymus europaeus* L., containing five acetate groups and evoninic acid. Additionally, it has been reported to exhibit various biological activities such as anti-viral, anti-cancer, anti-trypanosoma, and immunosuppressive activities [38]. Acanfolioside belonging to the Celastraceae family, as well as Evonine, have been reported to have biological activities such as anti-cancer and immunosuppressive activities [15]. Although Evonine and Acanfolioside have been detected as active substances in colon cancer cells, their efficacy on diseases and their potential as therapeutic candidates require further investigation. Future studies may explore the potential of these compounds in the treatment of colon cancer or other diseases. Since this study proved that ES is a potential therapeutic candidate for colon cancer, further research is needed on Evonine and Acanfolioside, the compounds of ES.

Finally, we cannot predict the side effects that will occur when ES is applied to humans, because we have confirmed the anti-cancer effect of ES only in vitro. However, when treated with ES by concentration (0, 12.5, 25, 50, 100 μg/mL) in CCD-18Co cells, which are normal colon epithelial cells, we found that they had more than 89% of cell viability at all concentrations, with a smaller decrease than in colon cancer cell lines. Therefore, considering these experimental results, it suggests that the dose and duration of administration can be determined during in vivo or clinical trials, and a positive effect can be expected even when ES is applied to humans.

## 4. Materials and Methods

### 4.1. Preparation of Euonymus Sachalinensis (F. Schmidt) Maxim. (Leaf) Extract

The plant extracts (KPM052-099) used in this study were obtained from the Korea Plant Extract Bank at the Korea Research Institute of Bioscience and Biotechnology, Daejeon, Republic of Korea. In addition, this plant was collected in Ganghwa-gun, Incheon, Republic of Korea, in 2019, and the voucher specimen (KRIB 0086779) was stored in the herbarium of the Korea Research Institute of Bioscience and Biotechnology. After drying in the shade, 100 g of the powdered plant was added to 1 L of 99.9% (HPLC grade) methyl alcohol and extracted at 40 kHz and 1500 W for 15 min and ultrasonication for 120 min. Under the condition of standing per cycle, the extraction was performed for 30 cycles at room temperature using an ultrasonic extractor. Then, 7.33 g of *E. sachalinensis* extract was obtained through filtration (Qualitative Filter No. 100, Hyundai Micro Co., Ltd., Seoul, Republic of Korea) and drying under reduced pressure.

### 4.2. Liquid Chromatography-Mass Spectrometry (LC-MS)

Samples for LC-MS analysis were prepared by dissolving at 1000 ppm using 99.9% methyl alcohol, according to the ES content information provided by the Korea Plant Extracts Bank (KPEB). Based on the study by An et al. [39], an ES component analysis was conducted using LC-MS analysis.

### 4.3. Cell Lines Culture

HCT116 cells with p53 wild-type characteristics, HCT116 cells with p53 null-type characteristics, and HT29, DLD-1 cells were obtained from the Korean Cell Line Bank (KCLB, Seoul, Republic of Korea). These human colon cancer cells were cultured in RPMI1640 medium containing 10% fetal bovine serum (FBS) and 1% antibiotics, and were maintained in an incubator at CO_2_ of 5% and 37 °C.

### 4.4. Cell Viability Test

The cytotoxicity of ES was measured using the 3-(4,5-dimethylthiazol-2-yl)-2,5-diphenyltetrazolium bromide (MTT) assay. HCT116 cells with p53 wild-type and p53 null-type characteristics and HT29, DLD-1 cells were seeded in 96-well plates (1 × 10^4^ cells/well) and ES were treated with each concentration for 24 h. Following treatment, MTT (2 mg/mL) was added to each well to have a final concentration of 0.5 mg/mL, and then, dimethyl sulfoxide (DMSO) was treated to dissolve formazan. A microplate reader (Molecular Devices Co., San Jose, CA, USA) was used to measure the optical density at 540 nm.

### 4.5. Western Blotting

Western blotting was performed on the HCT116 with p53 wild-type characteristics and HT29 cells to analyze the protein levels. Briefly, cells were seeded in 6-well plates (2 × 10^5^ cells/well) and treated with ES. The cells were then harvested using phosphate-buffered saline (PBS) and lysed using lysis buffer (Cell Signaling Technology, Dallas, TX, USA). The protein concentration was measured using the Bradford assay. Proteins (20 µg) were loaded onto 12% polyacrylamide gels for electrophoresis. The protein marker was purchased from GENETBIO (Daejeon, Republic of Korea). When the proteins were transferred to a nitrocellulose(NC) membrane, the membrane was blocked for 1 h using Tris-buffered saline Tween-20 (TBS-T) containing 3% skim milk. Next, the membrane was incubated with the primary antibody overnight at 4 °C, and the secondary antibody was incubated at room temperature the next day. Protein expression levels were detected using chemiluminescence (ECL) luminescent solutions. A detailed description of the antibodies is provided below: MID1IP1 (Cat No.15764-1) was purchased from ProteinTech Antibody Group (Chicago, IL, USA); PARP (Cat No. 9542S), CNOT2 (Cat No. 34214), and Cleaved-Caspase 3 (Cat No. 9661S) were purchased from Cell Signaling Technology (Dallas, TX, USA); and c-Myc (Cat No. 32072) was purchased from Abcam (Cambridge, UK), which were detected after binding to secondary antibodies (anti-rabbit, 1:10,000). Furthermore, Caspase 3 (Cat No. sc7272), p53 (Cat. sc-126), and β-actin (Cat. sc-47778) were purchased from Santa Cruz Biotechnology (Santa Cruz, CA, USA), which were detected after binding to secondary antibodies (anti-mouse, 1: 10,000).

### 4.6. Cycloheximide (CHX) Assay

HCT116 and HT29 cells were seeded in a 6-well plate (2 × 10^5^ cells/well) and treated with 100 µg/mL ES for 24 h. Afterward, the cells were treated with 50 µg/mL CHX at each time point (0, 15, 30, and 60 min), and protein levels of c-Myc and β-actin were confirmed using Western blotting.

### 4.7. Manufacture of 5-FU-R-HCT116 Cells

Parental HCT116 cells were treated with 5-FU at 0, 0.01, 0.1, 0.5, 2, and 10 µM to gradually make the cells resistant to 5-FU.

### 4.8. IF (Immunofluorescence)

HCT116 and HT29 cells were seeded in a 4-well confocal slide (5 × 10^4^ cells/well) and were treated with ES (100 µg/mL) for 24 h. Then, cells were enrooted for 15 min in 4% paraformaldehyde and washed with PBS three times. Cells were incubated with 0.2% Triton X in PBS for 15 min and washed three times with PBS. The cells were then blocked with 3% BSA in PBS for 1 h and washed with PBS. Next, the cells were incubated with the primary antibody (c-Myc) and secondary antibody. Then, the nuclei of the cells were stained with 4′,6-diamidino-2-phenylindole (DAPI), and observed using the CELENATM S Digital Imaging System (Logos Biosystems, Inc., Anyang-si, Gyeonggi-do, Republic of Korea).

### 4.9. Terminal Deoxynucleotidyl Transferase Nick-End-Labeling (TUNEL) Assay

We determined that ES caused apoptosis in colon cancer cells using the DeadEnd™ Fluorometric TUNEL system kit (Promega, Madison, WI, USA). All colon cancer cells were seeded in 4-well confocal slides (5 × 10^4^ cells/well) and treated with 100 μg/mL ES for 24 h. Then, the cells were fixed for 15 min in 4% paraformaldehyde, washed with PBS twice, and incubated with 0.2% Triton-X in PBS for 5 min. Following further washing with PBS, the cells were incubated with equilibration buffer for 10 min and the TUNEL reaction mixture containing rTdT enzyme for 1 h. The cells were then stained with DAPI for 5 min, the mounting solution was dropped, and the cells were visualized using the EVOSR cell imaging system (Thermo Fisher Scientific, Waltham, MA, USA).

### 4.10. Serum Stimulation

Colon cancer cells (HCT116 and HT29) were seeded in a 6-well plate at a density of 2 × 10^5^ cells/well and treated with 100 μg/mL ES for 24 h the next day. HCT116 cells treated with ES (100 µg/mL) were starved in 0.2% FBS for 24 h and then stimulated with 20% FBS for different periods (0, 6, 12, and 24 h).

### 4.11. Statistical Analysis

Data were analyzed and graphed using GraphPad Prism 8.0 (San Diego, CA, USA). A *t*-test (Mann–Whitney test) was used to compare the differences between groups and quantify this difference. A one-way analysis of variance (Tukey’s test) was used to perform all pairwise comparisons between groups using studentized range statistics. The graphs included present the mean ± standard deviation. Statistical significance was set at a *p*-value of < 0.05. In addition, all significance indications above the column graphs are from the data obtained from three or more experiments.

## 5. Conclusions

Our study confirmed that ES induced apoptosis in colon cancer cell lines. In particular, it was confirmed that apoptosis was caused by inhibiting the expression of c-Myc, and it was also shown that apoptotic factors such as PARP and Caspase 3 and oncogenes such as CNOT2 and MID1IP1 were controlled. Therefore, ES is a potential drug with anti-cancer effects.

## Figures and Tables

**Figure 1 molecules-28-03473-f001:**
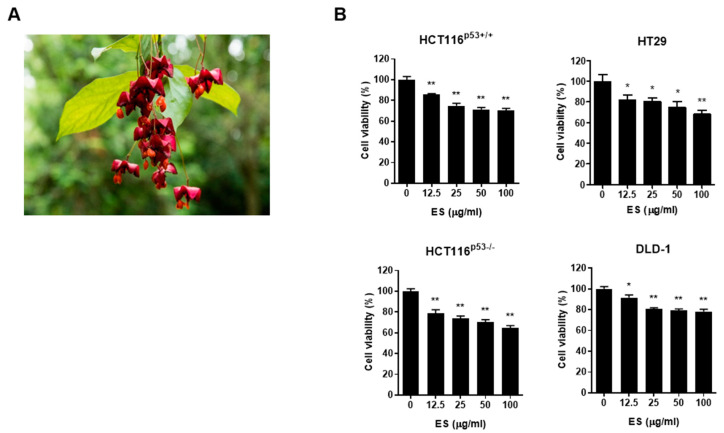
ES inhibits cell growth in colon cancer cells. (**A**) ES. (**B**) Effects of ES in colon cancer cells. Cell viability was confirmed when ES was treated for 24 h in colon cancer cells by MTT assay. This experimental data demonstrates that when treated with ES, colon cancer cells inhibited cell viability in a significantly more dose-dependent manner (0, 12.5, 25, 50, 100 μg/mL). All data are presented as the mean ± SEM. *n* = 5–6. * *p* < 0.05, ** *p* < 0.005 with the control group.

**Figure 2 molecules-28-03473-f002:**
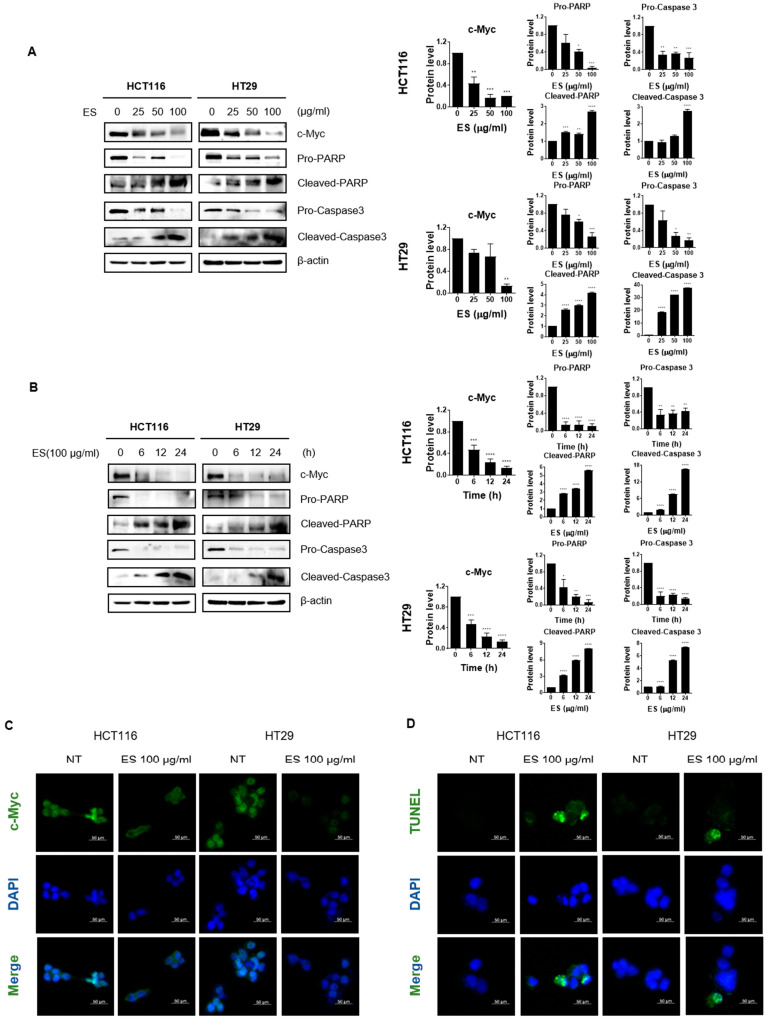
Effect of ES c-Myc and apoptosis-related proteins in colon cancer cells. (**A**) Colon cancer cells were treated with ES at each concentration (0, 25, 50, 100 µg/mL) for 24 h. The expression levels of c-Myc and apoptosis-related proteins, including pro-PARP, cleaved-PARP, pro-Caspase 3, and cleaved-Caspase 3, were confirmed in the cells by Western blotting. Additionally, the expression level of β-actin, which is used as an internal control, was also checked to confirm whether the equivalent protein was loaded. (**B**) Colon cancer cells were treated with ES 100 µg/mL at different times (0, 6, 12, and 24 h). The expression levels of c-Myc and apoptosis-related proteins, including pro-PARP, pro-Caspase 3, and β-actin, were detected in the colon cancer cells by Western blotting. (**C**) IF data show that ES treatment decreased the nuclear c-Myc level in the colon cancer cells. (**D**) TUNEL/DAPI staining shows ES induced apoptosis in the colon cancer cells. TUNEL positive cells were stained green, and nuclei were stained blue with DAPI. Taken together, we demonstrated that ES induces apoptosis in colon cancer cells. The magnification of every image is 200×. Scale bar, 200 μm. All data are presented as the mean ± SEM. *n* = 3–4. * *p* < 0.05, ** *p* < 0.005 and *** *p* < 0.001, **** *p* < 0.0001 with the control group.

**Figure 3 molecules-28-03473-f003:**
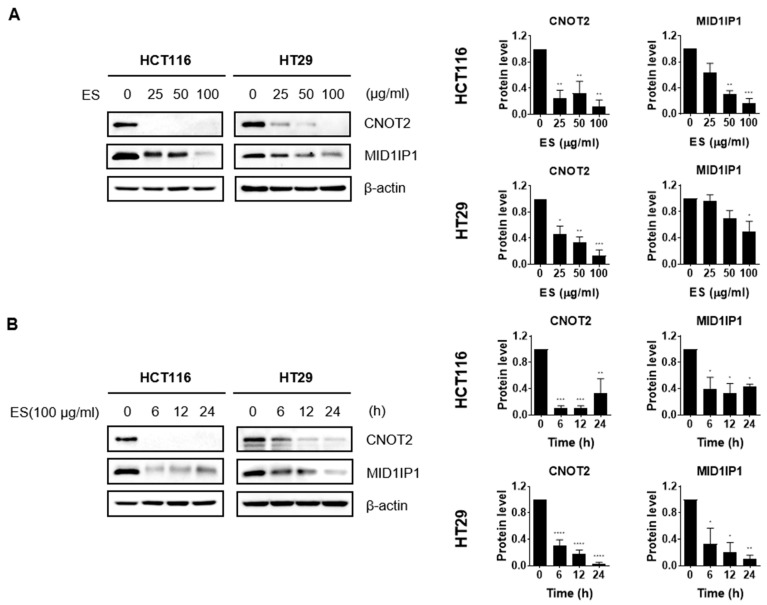
Effect of ES oncogenes in colon cancer cells. (**A**) The colon cancer cells were treated with each concentration (0, 25, 50, and 100 µg/mL) of ES. (**B**) The cells were treated with ES 100 µg/mL at different times (0, 6, 12, and 24 h). Using Image J, graphs were created that compare the expression levels of the housekeeper β-actin and each primary antibody. These results suggest that ES inhibits the expression of oncogenes, including CNOT2 and MID1IP1, in colon cancer cells. All data are presented as the mean ± SEM. *n* = 3–4. * *p* < 0.05, ** *p* < 0.005 and *** *p* < 0.001, **** *p* < 0.0001 with the control group.

**Figure 4 molecules-28-03473-f004:**
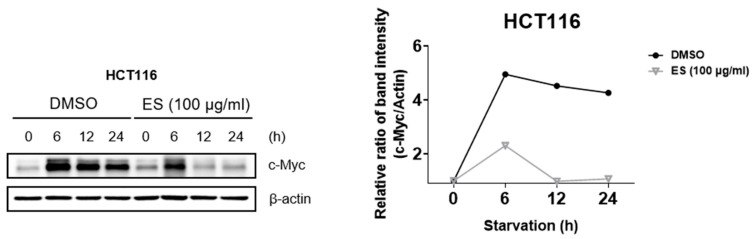
Effect of ES expression of serum-stimulated c-Myc in HCT116 cells. HCT116 cells treated with ES (100 µg/mL) were starved in 0.2% FBS and were serum-stimulated (with 20% FBS). Afterward, when the expression of c-Myc was confirmed, it was found that the ES was regulated.

**Figure 5 molecules-28-03473-f005:**
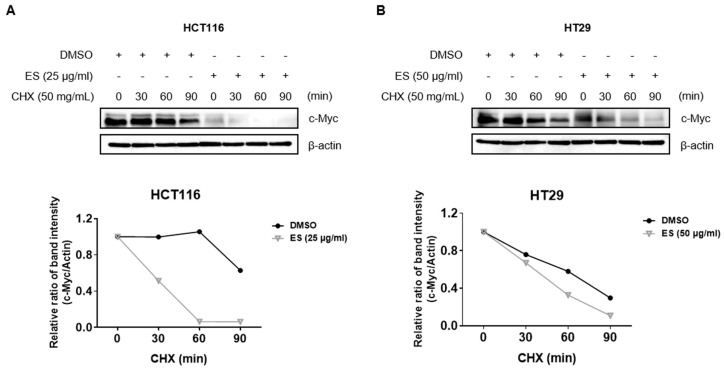
Effect of ES c-Myc stability in colon cancer cells. (**A**) HCT116 cells were treated with ES (25 µg/mL) for 24 h. Then, the cells were exposed to 50 µg/mL CHX at different time points (0, 30, 60, and 90 min). (**B**) HT29 cells were treated with ES (25 µg/mL) for 24 h. Then, the cells were exposed to 50 µg/mL CHX at different time points (0, 30, 60, and 90 min).

**Figure 6 molecules-28-03473-f006:**
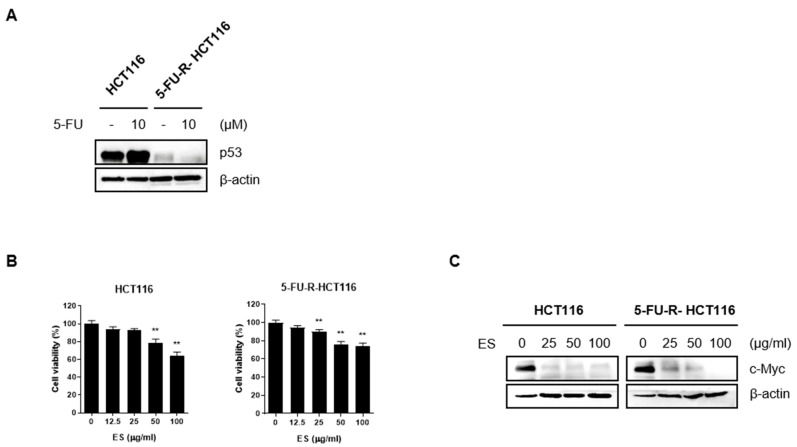
ES inhibits c-Myc expression in normal colon cancer cells and colon cells resistant to 5-FU. (**A**) After making cells resistant to 5-FU, we confirmed that the HCT116 cells were resistant to 5-FU by checking the expression level of p53 using Western blotting. In normal HCT116 cells, p53 expression was increased, whereas the p53 expression level did not change in 5-FU-R-HCT116 cells. (**B**,**C**) When ES was treated with different concentrations, cell viability and the expression of c-Myc were decreased in a dose-dependent manner in both general HCT116 cells and HCT116 cells resistant to 5-FU. This demonstrated that ES was also effective in 5-FU-R-HCT116 cells. All data are presented as the mean ± SEM. *n* = 6. ** *p* < 0.005 with the control group.

**Figure 7 molecules-28-03473-f007:**
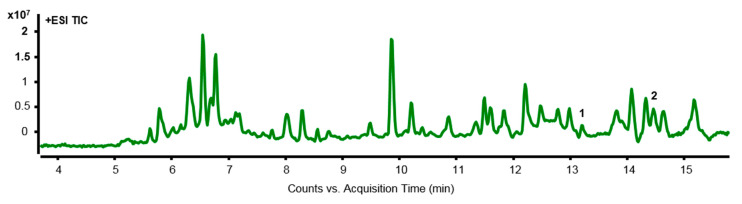
LC-MS chromatography of ES methanol extract. The overall chromatogram of ES is shown, and the peaks of compounds that have the potential to act as active substances are numbered. The names of compounds corresponding to each peak number are as follows; 1: Evonine, 2: Acanfolioside.

**Figure 8 molecules-28-03473-f008:**
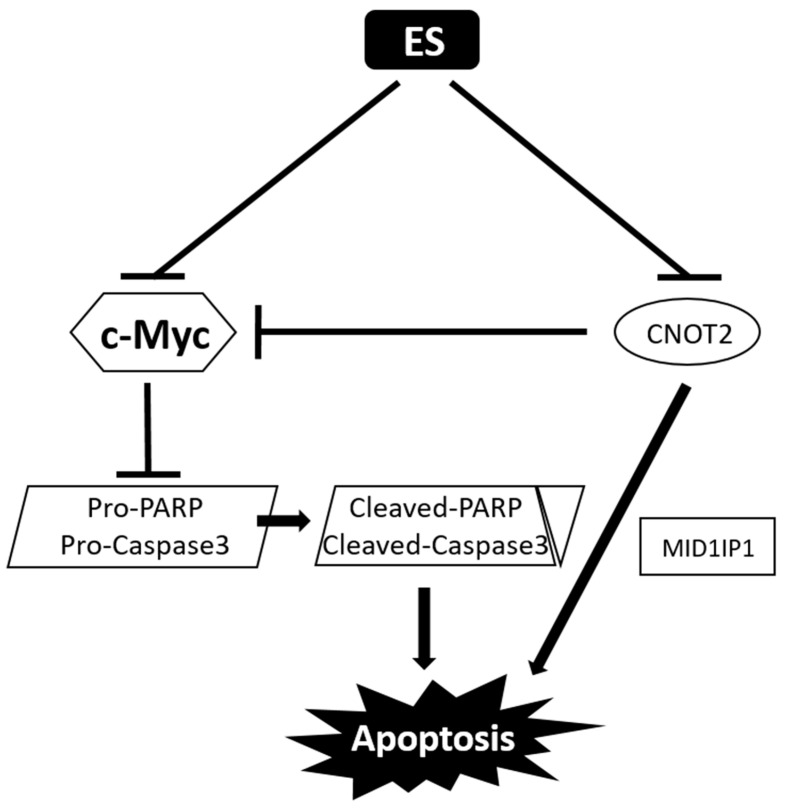
Mechanism scheme of ES in colon cancer cells. ES suppresses both c-Myc and CNOT2 in colon cancer cells. CNOT2 decreases c-Myc; pro-PARP and pro-Caspase3 are downregulated; and cleaved-PARP and cleaved-caspase3 are upregulated, leading to apoptosis. In addition, inhibition of CNOT2 induced apoptosis via MID1IP1.

**Table 1 molecules-28-03473-t001:** List of compounds detected in ES Methanol extracts using LC-MS.

No.	Compound	R.T (min)	Mass	Molecular Formula	Experimental Mass (*m*/*z*)	Selected Ion Species
1	Evonine	13.228	773.2531	C36H43NO17	(+) 779.2693	(M+H_2_O)+
2	Acanfolioside	14.469	762.2735	C37H46O17	(+) 801.3478	(M+K)+

## Data Availability

Not applicable.

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
