# Peer review of "Euonymus sachalinensis Induces Apoptosis by Inhibiting the Expression of c-Myc in Colon Cancer Cells"

_molecules, 2023, doi:10.3390/molecules28083473_

Round 1

Reviewer 1 Report

This is an article on an interesting and not well-studied topic. I appreciate the depth of research you have conducted on the topic. However, I suggest that you focus more on the key points that you want to convey in your paper. It would be helpful to provide more context and background information for readers who may not be familiar with the topic. Also, it would be useful to further develop your arguments and provide more evidence to support your claim.

1.     In line-91, author mentioned number of cells decreased on a treatment. But author preformed MTT assay it represent cell metabolic state. Author need to explain clearly the results.

2.     In figure 3, the expression of MID1IP1 expression was higher in HT29 cell line in 24 hour, author can give scientific explanation for this.

3.     Figure 4 and figure 5 is same, can author explain which figure is repeated and provide exact figure for the results.

4.     Author can explain results and discussion in detail. 

Author Response

Reviewer 1

This is an article on an interesting and not well-studied topic. I appreciate the depth of research you have conducted on the topic. However, I suggest that you focus more on the key points that you want to convey in your paper. It would be helpful to provide more context and background information for readers who may not be familiar with the topic. Also, it would be useful to further develop your arguments and provide more evidence to support your claim.

Point 1: In line-91, author mentioned number of cells decreased on a treatment. But author preformed MTT assay it represent cell metabolic state. Author need to explain clearly the results.

(Response) As you pointed out, I changed the content that I evaluated the cytotoxicity of cells through MTT assay. Thank you for pointing out to clarify the results of the experiment.

(Line 96-100)

Point 2: In figure 3, the expression of MID1IP1 expression was higher in HT29 cell line in 24 hour, author can give scientific explanation for this.

(Response) As the reviewer pointed out, we looked at Figure 3 and determined that a retest was necessary. So I made a new sample and checked it through western blotting. As a result, it was confirmed that the expression of MID1IP1 and CNOT2 decreased time-dependent when ES was treated in HT29 cells, so Fig 3 was changed.

(Line 138, 139)

Point 3: Figure 4 and figure 5 is same, can author explain which figure is repeated and provide exact figure for the results.

(Response) Thank you for your opinion so that the quality of the paper can be improved. We corrected it by confirming that the picture in Figure 4 is attached repeatedly in the place of Figure 5. Figure 5 is shown below.

(Line 166, 167)

Point 4: Author can explain results and discussion in detail.

(Response) As you pointed out, As you pointed out, in the Results section, we added the names of the cells used and the description of how the experiment was conducted. In the discussion section, we add an explanation for the lack of explanation, such as the perspective of whether ES can be applied to humans and the information in the active components of ES. In addition, we added the limitations and strengths of this study with critical insight.

(Results : Line 97, 162, 163)

(Discussion : Line 222-225, 269-278, 281-288)

Reviewer 2 Report

The paper by Park et al. entitled “Euonymus sachalinensis induces apoptosis by inhibiting the expression of c-Myc in colon cancer cells” is a research article on the effect of Euonymus sachalinensis (ES) on colon cancer cell. The authors performed MTT assays, immunocytochemistry, PCR, western blotting, and LC-MS analysis using two colon cancer cell lines, and found that ES induced apoptosis by inhibiting c-Myc. I would like to raise the following concerns.

1, The volume of experiments and obtained data may be unsatisfactory. All the results came from only two cell lines, without any in vivo study. I would like to strongly encourage the authors to perform additional in vivo study to confirm the conclusions. All the in vitro studies should be performed with additional colon cancer cell lines and normal control cell.

2, How did the concentration of ES was determined? Please provide the rationale. In addition, the authors should examine the toxicity of ES in normal control cell.

3, As the authors mentioned in the final paragraph in the Discussion, further research on the compounds of ES (Evonine and Acanfolioside) is needed in this paper, not for the future research.

Author Response

Reviewer 2

The paper by Park et al. entitled “Euonymus sachalinensis induces apoptosis by inhibiting the expression of c-Myc in colon cancer cells” is a research article on the effect of Euonymus sachalinensis (ES) on colon cancer cell. The authors performed MTT assays, immunocytochemistry, PCR, western blotting, and LC-MS analysis using two colon cancer cell lines, and found that ES induced apoptosis by inhibiting c-Myc. I would like to raise the following concerns.

Point 1: The volume of experiments and obtained data may be unsatisfactory. All the results came from only two cell lines, without any in vivo study. I would like to strongly encourage the authors to perform additional in vivo study to confirm the conclusions. All the in vitro studies should be performed with additional colon cancer cell lines and normal control cell.

(Response) In vivo experiments on ES in Colon Cancer are being planned for further study. By conducting MTT assays using HCT116 cells with p53 null-type characteristics and DLD-1 cells in vitro, additional cytotoxicity of ES was confirmed in colon cancer cells (Line 97, 100, 101, 310, 311, 318, 319), and similarly tested using CCD-18Co as normal control cells. As a result, it was confirmed that when ES was processed, the decrease in cell viability was found in colon cancer cells in a dose-dependent manner. It was also confirmed that normal cell lines have higher cell viability than colon cancer cell lines. However, when treated with 50 μg/mL of ES in CCD-18Co cells, we found a significant decrease in cell viability of 89% or more, and found that it decreased less than in colon cancer cell lines. (It was written in Line 282-288)

This data suggests that in vivo or clinical trials, dose and duration can be determined.

The data was obtained from at least three independent experiments and shown as means ± standard errors of the means. *p < 0.05 and **p < 0.01 compared to the control (non-treated).

Point 2: How did the concentration of ES was determined? Please provide the rationale. In addition, the authors should examine the toxicity of ES in normal control cell.

(Response) Based on the result of MTT assay, Western blotting was conducted. As a result, changes in apoptosis-related factors or oncogenes, as well as c-Myc, were evident in the two colon cancer cell lines at a concentration of 100 μg/mL of ES. So we set the concentration of ES to the highest concentration of 100 μg/mL. To show a definite change, we conducted IF, TUNEL assay, and serum simulation experiments at the high concentration we set. Also, in the case of CHX, the concentration was set when the expression of c-Myc in each cell line was reduced by more than 50% by referring to the Western blotting results treated with ES by concentration. Cytotoxicity in the normal control cell was confirmed through MTT assay using CCD-18Co cell and mentioned the relevant matters in point 1.

(Line 282-288)

Point 3: As the authors mentioned in the final paragraph in the Discussion, further research on the compounds of ES (Evonine and Acanfolioside) is needed in this paper, not for the future research.

(Response) Thank you for your opinion to improve the quality of the paper. However, the main topic of this study is extract, so the anti-cancer effect on the compound on the extract is planned as further study. However, as you pointed out, I thought that information about Evonine and Acanfolioside, which are active compounds of ES, should be supplemented, so I added the contents to Discussion.

(Line 269-278).

Reviewer 3 Report

The authors aimed to evaluate whether ES was effective in the treatment of colon cancer by treating a colon cancer cell line with ES and checking the expression change of c-Myc.

- The study is novel and fills a gap in the current literature.

- Methods are appropriate.

- Results are very detailed.

- Discussion section is too superficial in the current format and needs more critical insight.

- English should be improved for style.

Author Response

Reviewer 3

The authors aimed to evaluate whether ES was effective in the treatment of colon cancer by treating a colon cancer cell line with ES and checking the expression change of c-Myc.

Point 1: The study is novel and fills a gap in the current literature.

(Response) We sincerely appreciate your positive opinion.

Point 2: Methods are appropriate.

(Response) We sincerely appreciate your positive opinion.

Point 3: Results are very detailed.

(Response) We sincerely appreciate your positive opinion.

Point 4: Discussion section is too superficial in the current format and needs more critical insight.

(Response) As you pointed out, in the discussion section, we add an explanation for the lack of explanation, such as the perspective of whether ES can be applied to humans and the information in the active components of ES. In addition, we added the limitations and strengths of this study with critical insight.

(Line 222-225, 269-278, 281-288)

Point 5: English should be improved for style.

(Response) Thank you for your opinion so that the quality of the paper can be improved. We modified it so that someone can read this paper more smoothly.

(Line 45-62, 258-263)

Reviewer 4 Report

I would like to congratulate the authors for the structure of the manuscript and all the research carried out. It is highly publishable. However, there are some concerns, in part important, so the review articles need revision, see below.

Introduction

·       Why is this study considered relevant?

·       I suggest you incorporate a little more information related to Euonymus sachalinensis in relation to other clinical uses in humans or animal models.

·       Which herbal supplements have similar properties to Euonymus sachalinensis?

·       describe potential side effects

Materials and Methods

·                 The methodology is perfectly described and carried out

Results

·                 The tables/figures and the text describing them do not require any input, it is the strongest part of this study.

Discussion

·       Although there is an excellent description of the studies and results, possible mechanisms related to the results described in the manuscript could be added.

·       Would Euonymus sachalinensis be applicable to humans?

·       What are its limitations and strengths?

·       What does this article contribute to, the authors should make their own assessment and include their own discussion of the results shown in the manuscript?

Conclusion

·        In the Conclusion section, state the most important outcome of your work. Do not simply summarize the points already made in the body — instead, interpret your findings at a higher level of abstraction. Show whether, or to what extent, you have succeeded in addressing the need stated in the Introduction (or objectives).

Author Response

Reviewer 4

I would like to congratulate the authors for the structure of the manuscript and all the research carried out. It is highly publishable. However, there are some concerns, in part important, so the review articles need revision, see below.

Point 1: Introduction

  • Why is this study considered relevant?
  • I suggest you incorporate a little more information related to Euonymus sachalinensis in relation to other clinical uses in humans or animal models.
  • Which herbal supplements have similar properties to Euonymus sachalinensis?
  • describe potential side effects

(Response) With reference to the contents you pointed out, we added contents such as the necessity of research on ES and herbs with similar characteristics to ES to the introduction section. We searched hard for related content, but there was a limit to adding the content due to the lack of research on Euonymus sachalinensis at the moment, so we would appreciate it if you consider it be considered.

(Line 45, 46, 53-62)

Point 2: Materials and Methods

  • The methodology is perfectly described and carried out

(Response) We sincerely appreciate your positive opinion.

Point 3: Results

  • The tables/figures and the text describing them do not require any input, it is the strongest part of this study.

(Response) We sincerely appreciate your positive opinion.

Point 4: Discussion

  • Although there is an excellent description of the studies and results, possible mechanisms related to the results described in the manuscript could be added.
  • Would Euonymus sachalinensis be applicable to humans?
  • What are its limitations and strengths?
  • What does this article contribute to, the authors should make their own assessment and include their own discussion of the results shown in the manuscript?

(Response) With reference to what you pointed out, in the discussion section, we gave an additional explanation for the lack of explanation, such as the perspective of whether ES can be applied to humans and the information in the active components of ES. In addition, we added the limitations and strengths of this study with critical insight.

(Line 222-225, 269-278, 281-288)

Point 5: Conclusion

  • In the Conclusion section, state the most important outcome of your work. Do not simply summarize the points already made in the body — instead, interpret your findings at a higher level of abstraction. Show whether, or to what extent, you have succeeded in addressing the need stated in the Introduction (or objectives).

(Response) As you pointed out, I wrote a conclusion with the content of ’Our study confirmed that ES induced apoptosis in colon cancer cell lines. In particular, it was confirmed that apoptosis was caused by inhibiting the expression of c-Myc, and it was also shown that apoptotic factors such as PARP and Caspase 3 and oncogenes such as CNOT2 and MID1IP1 were controlled. Therefore, ES is a potential drug with anti-cancer effects.’. Thank you for your detailed opinion.

(Line 392-397)

Round 2

Reviewer 1 Report

The author has provided detailed explanation for the queries

Reviewer 2 Report

Thank you for the revised version of the manuscript. I have no additional comment.